# An Evaluation of the Implementation of a UK School-Based Running Program

**DOI:** 10.3390/children7100151

**Published:** 2020-09-25

**Authors:** Anna E. Chalkley, Ash C. Routen, Jo P. Harris, Lorraine A. Cale, Trish Gorely, Lauren B. Sherar

**Affiliations:** 1School of Sport, Exercise and Health Sciences, Loughborough University, Loughborough LE113TU, UK; j.p.harris@lboro.ac.uk (J.P.H.); l.a.cale@lboro.ac.uk (L.A.C.); l.b.sherar@lboro.ac.uk (L.B.S.); 2Diabetes Research Centre, Leicester General Hospital, University of Leicester, Leicester LE5 4PW, UK; ar516@leicester.ac.uk; 3Department of Physical Education and Sport Sciences, University of Limerick, Co Limerick V94 T9PX, Ireland; 4Department of Nursing and Midwifery, University of the Highlands and Islands, Inverness IV3 5SQ, UK; trish.gorely@uhi.ac.uk

**Keywords:** physical activity, program evaluation, implementation, school children, running

## Abstract

The adoption of school-based running programs has rapidly increased over the last five years in the UK and globally. However, there is currently a lack of information on how these initiatives are implemented, and whether they are generalizable and/or sustainable. This study evaluated the implementation (including reach, fidelity, and dose) of a school-based running program over seven months to inform future delivery. This observational study used a mixed-method, single-group, before-and-after design strengthened by multiple interim measurements to evaluate the implementation of an optional school-based running program. Five state-funded primary schools in Leicestershire, UK, participated, with 17 teachers and 189 (81 boys (47.4%) and 90 girls (52.6%)) Year 5 pupils (aged 9–10 years) from eight classes. During the 2016/2017 academic year, data were collected via several measures (including interviews, focus groups, observations, questionnaires, and teacher implementation logs) at multiple levels (i.e., school and individual) and at multiple time points during implementation. Follow up qualitative data were also collected during 2017/2018. The school-based running program achieved good reach, with 100% of pupils opting to participate at some point during the academic year. All schools implemented the program with good fidelity, although the level of implementation varied between schools and over time. The average number of sessions held per week ranged from 0.94–3.89 with the average distance accumulated per pupil per week ranging from 0.02 to 2.91 kilometers and boys being more likely than girls to be classed as high-level participators. Despite an initial drop off in participation over time, all schools remained engaged in the program and continued to implement it until the end of the school year. Contextual features (e.g., staff capacity and resources) differed between schools and influenced the quality of implementation and the frequency of delivery. The school-based running program is simple, inexpensive, and versatile and can be implemented by schools with relative ease. However, schools are diverse settings, with unique challenges to ongoing delivery. Thus, planned adaptations, specific to each school’s context, are likely necessary to sustain participation in the longer term and should be considered prior to implementation.

## 1. Introduction

School-based running programs (sometimes referred to as active mile initiatives) typically encourage children to be active during the school day by providing opportunities for pupils to walk and/or run around a marked route in the school grounds for a period of time (e.g., 15 min) [1]. Grassroots support for school-based running programs has rapidly increased in recent years [2] and many are being implemented as a pragmatic ‘solution’ to rising levels of childhood obesity, poor levels of fitness, and inactivity [3]. Indeed, the most well-known example, The Daily Mile^TM^ [4], is reported as being implemented in over 70 countries worldwide, and over 5867 registered schools have participated in the initiative in England alone [5].

In the United Kingdom (UK), political cross-sector commitment for running programs was reinforced by the 2019 cross-government School Sport and Physical Activity Action Plan [6], which advocated the use of active mile initiatives to establish physical activity as an integral part of the school day. However, evidence for the effectiveness of school-based interventions is mixed [7] and there are limited generalizable, effective, and sustainable physical activity interventions that have been translated into practice [8]. For example, a recent systematic review with meta-analysis found that school-based interventions had a moderate effect on physical activity during the school day, but this was not reflected in increases in physical activity across the whole day [9]. Possible explanations are that intervention components were not effective and/or they were not implemented appropriately. Consequently, a greater focus on documenting real-world implementation and fidelity as part of evaluations of public health interventions [10], school-based physical activity interventions [11], and school-based running programs [12] is needed.

While intuitively running programs are an appealing investment for schools [13], robust efficacy and implementation evidence is scarce [14]. In order to maximize their potential, there is a need to understand which pupils participate in these programs, if and how programs can be successfully transferred from one context to another, and the factors that may constrain or enhance the implementation process [15]. This study, therefore, examined the implementation of a school-based running program over seven months to understand the context within which it as implemented in each of the schools and the key elements of implementation including reach, fidelity, and dose (delivered and received).

## 2. Method

### 2.1. Programe Description

Marathon Kids (MK) is a school-based running program managed and delivered by the charity Kids Run Free (KRF). It is described in detail elsewhere [12,16,17], but, in brief, MK gives primary school pupils the opportunity to walk and/or run the distance of a marathon over the school year. The program aims to empower children of all fitness and ability levels to engage in regular physical activity. Pupils are encouraged to walk and/or run laps of a course on the school grounds once or twice a week. For every lap completed, pupils are given a lap band and the total number of bands collected per child is recorded via a digital tracking system (DTS). The DTS provides summary data regarding when pupils participate, how many laps each pupil completes, and their accumulated distance and rewards are available upon reaching key milestones, for example, quarter, half, three-quarter, and full marathon. Optional on-site support is also provided during a school launch event by a member of staff from KRF who, for example, delivers a marathon-themed assembly, demarcates the running route(s), and helps to set up and provide training on the administration of the program to the MK Champion (member of staff responsible for implementing MK) and a selection of MK Ambassadors (peer leaders). A summary of the program is provided in Appendix A (Table A1) using the Template for Intervention Description and Replication in Public Health Programmes (TiDier-PHP) [18].

### 2.2. Study Design

This study used a mixed-method, single-group, before-and-after quasi-experimental design. This design was chosen as a pragmatic approach to exploring the implementation process and informing the development and effectiveness of implementation strategies (as opposed to the effectiveness of the intervention itself) in a way that would have relevance to practitioners and policy makers. Furthermore, it was believed to yield naturally occurring differences in types or intensities of implementation strategies, as determined by the schools and their local contexts, in order to evaluate their influence on implementation [19].

### 2.3. Study Population and Recruitment

Schools from the East Midlands region of the UK were recruited as part of a partnership with a local Teaching School Alliance. (Teaching School Alliances are groups of outstanding schools that work with others to provide high-quality training and support for school improvement in their local area.) At a headteacher meeting facilitated by the Alliance, schools were offered the opportunity to participate in the study, following which 32 (representing a reach of 89.9%) expressed interest in being involved. The schools were categorized into tertiles based on the Income Deprivation Affecting Children Index (IDACI) [20]. They were then further stratified by geographic location (Urban, Rural - town and fringe, Rural - village), identified by the Edubase database (Edubase) [21], and school size (small, medium, large), according to the number of pupils on roll (from the Office for Standards in Education, Children’s Services and Skills (Ofsted) Data Dashboard, which subsequently shut down in September 2016). From the 32 schools, two from each tertile were invited to participate while also ensuring a spread of schools in terms of size and geographic location. In total, 12 schools were contacted and invited to participate in the study. On receiving further information, three provided no information with regards to not proceeding in the study, two stated lack of capacity, one had a recent change in staffing prohibiting them from participating, and one was unable to commit due to an unscheduled inspection. Consequently, five schools were recruited to the study, representing a reach of 41.7%.

The evaluation focused on Year 5 (9–10 years) children, although schools were encouraged to implement MK in all year groups. Given that the study was concerned with the school level and individual level characteristics associated with implementation, participants from each school included the headteachers and/or deputy headteachers, Year 5 pupils/teacher(s), and the MK Champion (if this was not the Year 5 teacher). Parental consent to participate was also obtained from the pupil’s parents/carers and each school provided written consent to participate in the study signed by the headteacher. All participants provided written, informed consent and verbal assent prior to participation in any data collection. The study has been approved by the Human Participants Ethics Subcommittee of Loughborough University (R16–P116).

### 2.4. Data Collection

During the 2016/2017 academic year, data were collected at five time points (TPs) over seven months (October 2016–June 2017) with follow-up data collected during the 2017/18 academic year at two TPs (Figure 1). Detailed methods for the evaluation of MK, including the theoretical framework adopted, have been published in elsewhere [12]. Table 1 outlines the implementation dimension assessed and the definition adopted, which was specific to MK.

#### 2.4.1. School Level Measures

##### Headteacher Interviews

Headteachers from each of the schools participated in a semi-structured interview to elicit contextual information pertaining to their school’s policies, practices, and ethos relating to the promotion, teaching, and delivery of physical activity and healthy lifestyles, as well as influences on the decision to adopt the program. See Appendix B (Table A2) for an example of thematic interview/focus group guide used for the evaluation.

#### 2.4.2. Individual Level Measures

##### Teacher Interviews

All Year 5 teachers and MK Champions (if different) participated in semi-structured face-to-face interviews. At TP2 and TP5, questions focused on: The characteristics of pupils and schools (participating and nonparticipating), the extent to which components of the intervention were used and completed, the delivery of MK and teacher satisfaction, pupil behavior and pupil uptake, and perceptions of influences on uptake. At TP6 and TP7, questions explored: The perceived long-term effects on pupils, teachers, and school(s); changes to policy and practice; and any factors affecting sustainability.

##### Teacher Log

To obtain data on the frequency and duration of exposure to MK as well as uptake, teachers were asked to complete a weekly survey (Survey Monkey, Palo Alto, California, USA) of each class’ participation during the implementation period (October 2016–June 2017). For example, data were collected on frequency, timing, and duration (in minutes) of use of MK during the week as well as contextual information such as surface/route used and supervision.

##### Pupils’ Demographics

At TP1, each pupil’s name, date of birth, sex, ethnicity, free school meal eligibility (eFSM), and home postcode were collected from the school information management system. Relative deprivation and area-level socioeconomic status were calculated based on home postcode using the 2015 English Index of Multiple Deprivation (IMD) [20]. Home postcodes were uploaded to an online ‘postcode lookup’ tool (http://imd-by-postcode.opendatacommunities.org/ accessed 3 November 2016), which outputted the corresponding IMD rank and decile.

##### Pupils’ Anthropometrics

Each pupil’s standing height and weight was measured and body mass index calculated [12] pre- (beginning of the school year) and post-program (end of the school year).

##### Pupils’ Physical Activity

Physical activity data were collected pre- and post-program with a waist-worn ActiGraph triaxial accelerometer (GT3X, GT3X+, or BT) (ActiGraph, Pensacola, FL, USA). Participants were instructed to wear the device on their right hip during all waking hours, except when engaging in water-based activities, for seven consecutive days.

##### Pupil Focus Groups

Semi-structured focus groups were conducted with pupils at TP2 and TP5. Focus groups were chosen to provide rich data relating to pupils’ perceptions of MK, facilitate a detailed description of the implementation of MK, understand how those perspectives may be different from, or similar to, their peers’ as well as their teachers’, and to contribute to data previously collected on pupils’ experiences of participating in MK [16]. One mixed-sex group of six to eight Year 5 pupils per class participated (*n* = 53 in total) and were selected by the MK Champions based on the following guidelines: Ensuring a range of pupils in relation to sex, enthusiasm, and participation level in the program (e.g., high, medium, and low), as well as their willingness to communicate experiences and opinions.

#### 2.4.3. Running Program Participation

At TP2, TP3, and TP4, pupils’ participation in MK during a lunchtime period was assessed using direct observation. Data collected during these structured observations were largely qualitative and involved the researcher making detailed notes of relevant contextual and other information (e.g., weather, route used, duration of the session, level and type of interactions observed (i.e., physical or verbal and prosocial or antisocial interaction), fidelity to the MK protocol, and approximate numbers of pupils participating). Participation in MK was ascertained via triangulating data from the DTS, direct observations, and the implementation log completed by teachers. Specifically, data from the DTS were verified using contextual information from the teacher implementation log to determine the dose (i.e., distance covered) for each child pupil over time.

### 2.5. Data Analyses

#### 2.5.1. Qualitative Data

All qualitative data were deductively coded, using a framework approach to thematic analysis [22], into a priori themes based on the implementation outcomes of interest (i.e., reach, fidelity, dose). The framework approach was chosen to help link data from different sources within the study and map the dynamics of implementation over time, enabling comparisons between, and associations within, cases to be identified systematically [23]. Firstly, once transcribed, the transcripts were read and reread in order to become familiar with the breadth and depth of content of the data and to generate preliminary ideas and notes for coding. For the purpose of data reduction, transcripts were ‘selectively coded’ whereby a corpus of instances in which implementation outcomes (a priori themes) were discussed, were identified in the data, and were chosen for coding. A deductive approach to analysis was taken, in which codes were collected under the a priori themes before comparing the coding clusters together and in relation to the entire data set. A framework matrix was generated in NVivo (QSR Version 12.0) with the column of the matrix representing the a priori themes and each row representing a participant. Data were ‘charted’ into the matrix by abstraction and synthesis. Each coded piece of text was distilled into a summary of the respondents’ views and entered into the matrix. The original text was referenced, allowing the source to be traced and the process of abstraction to be examined and replicated. Care was taken to reduce the data while retaining the original meanings. Analytic memos were produced to capture impressions and early interpretations of the data and provide an audit trail of the research process, thus enhancing the rigor of the analysis [24]. Once all data was charted, they were interpreted as a whole by moving back and forth between cases. Within-case analysis was employed to highlight similarities and discrepancies between data sources, in combination with cross-case analysis, to identify patterns across cases in order to map the range of experiences of implementation and find associations, patterning, and structure within the data.

#### 2.5.2. Quantitative Data

Descriptive statistics, expressed as mean and standard deviation (SD) or percentages, were employed to explore the school, teacher, and pupil level characteristics, as well as level of implementation (obtained from teacher logs and DTS) over time and between schools.

## 3. Results

Overall, five schools, 17 teachers, and 189 pupils from eight classes were recruited to the study. The final sample consisted of 16 teachers and 171 pupils (81 boys (47.4%) and 90 girls (52.6%)).

The schools ranged in denomination, size, location, deprivation (IDACI and eFSM), and performance (Ofsted rating). (Ofsted, The Office for Standards in Education, Children’s Services and Skills, is a nonministerial department of the UK government, reporting to Parliament. Ofsted is responsible for inspecting a range of educational institutions. Ofsted ratings are the means by which Ofsted inspectors indicate the quality of an institution following an inspection. There are four ratings where ‘Outstanding’ is the highest, through to ‘Inadequate’, which is the lowest.) Schools 2 and 4 were federated (i.e., with one single overarching governing body accountable for both). The characteristics of each school are summarized in Table 2.

Five head/deputy heads, two teachers, and one teaching assistant were recruited for the interviews/focus groups. Of the recruited teachers, five adopted the role of MK Champions, three of whom were male, with an average of 10.2 ± 6.3 years of teaching experience, (4 ± 2.1 years in their current school).

The majority (73.9%) were White British and 26.1% of participants were classified as overweight or obese (Table 3). On average, pupils spent 58.9 (± 23.1) minutes per day in moderate to vigorous physical activity (MVPA) with 38.8% meeting the UK Chief Medical Officer’s physical activity guidelines [25].

The duration of the focus groups ranged between 28–63 min (average 42.5 min) and the interviews lasted between 12–43 min (average 23.4 min).

### 3.1. School Implementation Context

Of the five schools, two (Schools 1 and 2) implemented MK across all year groups; Schools 3 and 4 implemented it with Key Stage 2 only (7–11 years), and School 5 implemented it with all year groups apart from Reception (4–5 years). Schools experienced several challenges and to varying degrees, relating, for example, to staffing and poor facilities and infrastructure, when planning to implement MK. Every school offered MK as a lunchtime activity, which provided additional benefits.
We are a school that has always had activities to do because lunchtime is when you have the most incidents. So, we do have structured activities most lunchtimes for everybody anyway; that does help.MK Champion, School 4 (TP5)

School 3 also delivered MK at other times including before school, during Physical Education (PE), at lunchtime, and after school. The MK Ambassadors oversaw delivery in Schools 1 and 5, whereas the lunchtime supervisors and the MK Champion took responsibility for delivery in Schools 2 and 4 and in School 3, respectively.

All schools delivered MK throughout the entire academic year, although the number of weeks of delivery varied according to the different start time points (10 October to 28 November). Consequently, the minimum distance per week to complete a marathon ranged from 1.51 km (School 1) to 2 km (School 3). The length of the premarked route ranged from 70.2 m (School 2) to 124.5 m (School 5) and the dominant surface used for MK across all schools was tarmac, with Schools 4 and 5 using this for 100% of sessions. Three of the schools (Schools 1, 2, and 4) used existing markings on the playground. School 3 used the perimeter of a teaching block, while School 5’s route partly followed the outline of the trim trail on its playground.

### 3.2. Reach

Marathon Kids reached all pupils in Year 5 from all schools at some point during the academic year, although the dose of MK fluctuated. The qualitative data showed that there was a perception among staff that pupils who continued to participate were those who were characteristically competitive and responded positively to challenges (such as completing a marathon). In some schools these were the more active and ‘sporty’ children, but two of the five MK Champions commented on their surprise at several children participating who did not usually join in optional physical activity at school. This suggests that the program appealed to an array of children and not just the already active.
Just looking purely from my class, yes, it’s those that are always engaged, always enthusiastic, or always try their best whatever you throw at them. They’re the ones who tend to do it. But, coupled with that, you’ve got those who, shall we say, the less sporty inclined so those who don’t necessarily take part because it’s too much effort.Year 5 teacher, School 5 (TP2)

### 3.3. Fidelity

To determine the extent to which MK was implemented as intended, the schools’ delivery of MK was assessed against the core principles of the program. In accordance with an approach previously used in school-based physical activity interventions [26], fidelity markers were either coded as not employed at all (0), partially employed (1), or fully employed (2), based on data from the DTS, observations, implementation log, and interviews and focus groups. The fidelity scores for each school are summarized in Table 4. While all schools scored similarly for fidelity to MK, variation by marker of fidelity was evident. Schools scored the lowest on adhering to the reward-based strategy and issuing rewards at key milestones.

### 3.4. Adaptation

Planned adaptations in implementation related to the issuing of rewards and certificates and to the logging of laps, both of which were recorded as having partial fidelity. Two MK Champions (Schools 1 and 2) created their own certificates, considering these to be more personalized and/or to better reflect the wider values of MK such as rewarding perseverance (e.g., most consistent runner) rather than just distance. Lap bands were used in all schools to record the number of laps run during a MK session; however, for pragmatic reasons, School 3 did not adopt these when implementing MK during PE to save time. Instead, each pupil was given a minimum of five laps and pupils were asked to report if they completed more than this.

### 3.5. Dose of MK Delivered

There was good compliance (87.6% ± 6.5%) by the MK Champions in completing the online implementation log, with one school (School 2) completing 100% of the logs. School 4 reported the highest number of sessions per week (3.9 ± 2.5) and School 5 the least (0.9 ± 0.7) (Table 5). The average duration of a MK session varied between schools with the shortest sessions (0–5 min) being when it was used as a warm-up before PE and the longer sessions (>30 min) taking place during lunchtime. This was corroborated by the lunchtime observations.

### 3.6. Dose of MK Received

In all schools, a decline in the distance covered was seen after the first week of implementation; however, thereafter the average distance remained fairly stable. Table 6 summarizes the dose of MK received by pupils based on the data obtained from the DTS. The average distance accumulated per child per week, based on the school-reported data, ranged from 0.02 km to 2.91 km. The greatest average distance per child per week was observed in School 2 with 1.34 km and the least in School 5 (Class B) at 0.47 km.

Pupils were classed as being either low, medium, or high with regards to participation using the interquartile ranges of the average distance walked and/or run per week per pupil, taken from the DTS. Across all schools, boys were more likely than girls to be classed as high participating (44% vs. 12%). Differences within classes of the same school were also apparent, particularly in School 3 where the average distance per child per week ranged from 0.73 to 2.91. Children in School 1 achieved the farthest total distance of 597.6 km, closely followed by children in School 3. School 3 also had the two highest performing classes in terms of average distance per pupil and the most pupils to have completed a marathon. Meanwhile School 2, had the greatest average distance per participant of 32.28 km with 100% of participants having completed a half marathon or more.

The qualitative data support the DTS data with overall participation ebbing and flowing as initial interest and enthusiasm for MK waned before finding ‘*its natural level*’. At the outset, some MK Champions reported a mixed response among pupils with some very enthusiastic and others less so, but across all schools, initial high participation in MK was followed by a gradual decline.
It started off the first couple of weeks with big mobs of children running, very, very keen and enthusiastic. The assembly did enthuse them and that went really well, and then gradually as weeks have gone on numbers have gone down and down to the sort of current point where we maybe have sort of 10 running at a lunchtime.MK Champion, School 5 (TP2)

There was also an expectation among school staff at the outset that MK may not have the same appeal and longevity for all pupils; however, they felt that this was typical with many initiatives introduced into school, particularly when pupils self-select to participate. Interestingly, MK Champions frequently commented on how they thought MK was more effective for particular year groups. For example, in School 1, Year 3 had the highest participation rate whereas in School 2, Year 4 had the most committed participators.

### 3.7. Level of Implementation

Schools were categorized as being high, medium, or low implementers based on a combination of the interquartile range of dose of MK delivered and fidelity of delivery (see Table 7). One school was classified as having a low level of implementation (School 5), three as medium (Schools 1, 2, and 3), and one as having a high level of implementation (School 4).

### 3.8. Quality of Implementation

The quality of delivery of MK by teachers was assessed via interviews with school staff, focus groups with pupils, and observations. Several themes were identified. Staff capacity and staff buy-in were felt to be the most important determinants of how well MK had been implemented and drove many of the decisions regarding delivery. Noticeably, the role of the MK Champion within each school was thought by some to be a significant factor relating to the effectiveness of the program’s implementation. This was not only in terms of their ability to engage pupils and encourage participation but also in terms of ensuring MK was manageable and ‘*working for the school*’. This was contextualized to the school, although there was an agreement from all that the role of the MK Champion was an essential element of the program.
I think the program needs someone who, someone in charge really because it would fall apart.MK Champion, School 1 (TP6)

For example, the MK Champion in School 1 was chosen for the role because it complimented her other responsibilities within the school, (i.e., as lead for healthy schools). In School 2, the deputy headteacher felt that her strategic role within the senior management team helped to gain staff buy-in and ensure that “*Marathon Kids became part and parcel of what we do as a school*”. The influence of the MK Champion was most evident in School 3, given the synergy between the nature and ethos of the program and his responsibility as PE lead, and in schools 4 and 5 by the absence of a MK Champion. Although MK Champions were allocated, individuals did not fulfill their roles because of lack of capacity (School 5) and insufficient clarity surrounding the role (School 4).

All teachers approved of their school’s participation in MK and were supportive of pupils taking part, agreeing that the anticipated health benefits were important and valued. However, all MK Champions and many teachers discussed lack of time as a limiting factor to the degree to which they and other members of staff had been able to engage with the program.
Teachers are very busy people and lunchtimes, in particular, are very busy. You’re trying to have your lunch, first of all, and then set up for the afternoon lessons. That’s what lunch hours tend to be. So, there was limited opportunity for us to go out there.MK Champion, School 5 (TP5)

Many pupils highlighted a lack of support from class teachers particularly as time progressed (e.g., in praising their participation and/or providing feedback on their progress). Some teachers, however, were active role models and firm advocates for the program. Most notably, this was seen in School 1 where both the headteacher and MK Champion participated alongside the pupils. Indeed, in this school, staff participation in MK was highlighted by all pupils as being an enjoyable element of the program and provided a rare opportunity to chat informally with the headteacher. It was also evident during the observations that the headteacher’s participation in MK was influential in setting a level of expectation for the school and establishing participation in the program as the norm.
We have some staff members who were ‘Well, why are we doing that for all the classes?’. I don’t have many negative Nellies. I’m hoping that because I’m going to go out there and do it, and [MK Champion] has said that she wants to, I’m hoping that we can bring them along with the tide.Headteacher, School 1 (pre-program)

In Schools 1, 3, and 5, delivery was predominantly facilitated by the MK pupil Ambassadors with varying degrees of support from the MK Champions and/or other members of staff. This was an efficient method of delivery in School 1, with the introduction of a roster system leading to a structured and organized running of the program. However, other schools relied almost entirely on the MK Ambassador input, due to limited school capacity to organize and champion the program and at a detriment to the quality of the program.

For all schools, the timing of MK was determined by logistical influences such as the capacity to supervise and manage the delivery of MK. School 4’s high level of implementation of MK was partly attributed to the fact that lunchtime supervisors coordinated delivery. Indeed, when observed, the MK sessions appeared to be an organized and consistent part of the school’s lunchtime routine.


*Most of the time now you don’t even know you’re doing it because you automatically just do it and you don’t know that you are just doing it yourself.*
Year 5 boy, School 4 (TP5)

School 2 (federated with School 4) used the same approach to delivering MK where the MK Champion also frequently referred to the program as becoming ‘*routine*’ and part of the ‘*status quo*’. However, participation in MK was equally perceived by many pupils from both schools as a compulsory activity whereby they were required to achieve a minimum number of laps before being ‘*allowed to play*’ and, indeed, the lunchtime supervisors were often observed providing instructions to pupils in an authoritarian manner rather than encouragement. While this style of implementation may have contributed to the high dose of MK, it may have been at the expense of pupils’ sense of enjoyment and autonomy of the activity, as one pupil stated: “*Sometimes it’s a bit of a torture because you just want to play.*” Year 5 boy, School 2 (TP5).

Conversely, in School 5, roles and responsibilities for delivering MK were not so clearly defined. Consequently, frequent disagreements were observed among the MK Ambassadors and between the MK Ambassadors and other pupils in the playground, leading to disruption in delivery. In this school, pupils felt that adult support was needed to deliver MK, specifically to help protect the integrity of the monitoring system (i.e., the perception that pupils lacked authority to deal with cheating and disruptive behavior). In hindsight, teachers also reflected on this challenge.
Yeah, in all honesty, I didn’t have the time to run it properly. What it really needed is a member of staff out there in a tracksuit and trainers every day running alongside the children, being that role model.MK Champion, School 5 (TP6)

The distal nature of most of the MK Champions to pupils’ participation in MK furthermore meant that there was often a delay in them receiving feedback on their progress and distance covered, if they received any at all. Many pupils described how this compromised their motivation to participate in MK.
We should get, like, a ‘Well done’ from our teachers. They don’t really say anything… I think it would encourage us to keep doing more and more bands. I think that would, like, encourage us.Year 5 girl, School 4 (TP5)

## 4. Discussion

This evaluation was conducted to determine the extent to which a school-based running program, MK, was implemented. Specifically, it sought to understand the key elements of implementation including reach, fidelity, and dose.

### 4.1. Reach

MK appeared to appeal equally to both the active and non-active pupils in the schools; however, boys were more likely to be classified as high participating than girls. This is, perhaps, not surprising given that, pre-program, a higher percentage of boys were meeting the physical activity recommendations, and gender differences in physical activity participation is one of the most pervasive findings in the literature [27]. Systematic gender differences in how school-based running programs are received has also recently been reported elsewhere [28]. Research further suggests potential differences in how much physical activity children gain from interventions, with a greater tendency for girls to (verbally and physically) interact together while participating [29]. This is significant as social factors are powerful motivations for children’s participation in physical activity, especially for girls [30] and this seems to be the case here. Previous research posits that physical activity could provide an appropriate context within which to facilitate the development of peer relationships [31] and that friendships are associated with the promotion of self-worth, positive attitudes toward physical activity, and an increased likelihood of continued participation [32]. Given the social nature of walking and/or running (with autonomy over how, e.g., in pairs and/or groups) and incorporating both participants and MK Ambassadors, MK appears to provide appropriate opportunities for this. It is recommended that schools encourage staff to participate and become role models for the initiative as well as encourage pupils to participate together and facilitate social cohesion.

### 4.2. Fidelity

All schools implemented MK with good fidelity, achieving 75% or above adherence to the program’s implementation strategies. The qualitative data confirms that this may be attributed to the simple nature of the program and suggests that it is easier to achieve good fidelity with simple programs such as MK, compared to more complex multicomponent programs that are susceptible to poor fidelity, particularly in real-world contexts [11]. There is also the added benefit of such programs requiring very little or no ongoing investment in infrastructure and resources, making it credible for large-scale dissemination and adoption [33]. While these findings suggest that more could be done to improve the fidelity of MK’s implementation, the appropriateness of 100% fidelity has been called into question [34,35]. Durlak (2015) [34] contended that there may be a minimum threshold for implementation fidelity at which desired program outcomes are obtained. Indeed, heterogeneity in implementation is expected. This is particularly so within a school setting where adaptation is a necessity for strategies that target one or more levels within the system that support the adoption and implementation of the intervention [36]. Planned adaptations to the implementation of MK were seen in the current study and are recommended (e.g., issuing of rewards and certificates and logging of laps); however, it is important that the underlying features of the program are maintained [37].

### 4.3. Dose Delivered

Although the program was delivered with high fidelity, the dose of MK delivered differed considerably between schools. Interestingly, the high frequency of sessions in some schools did not translate to distance accumulated. Recent literature comparing the implementation of a school-based run/walk program and a classroom-based physical activity break reported that the run/walk program offered a higher dose of physical activity programming [26]. The variation could be partially accounted for by the delivery of MK at different times of the day and, hence, the degree of autonomy they allow [16], as well as the variability in influence of the MK Champion. The presence of a champion (i.e., an individual with sufficient influence and autonomy to galvanize the whole school and secure commitment from staff and pupils) has been demonstrated to be a predictor of school physical activity program implementation [38] and to influence pupils’ participation [16]. Indeed, Champion-led programs have advantages with respect to their acceptability, feasibility, and sustainability [39] and it is possible that their involvement, or lack of it, within the participating schools is responsible for some of the effect on dose (delivered and received) within the current study. For meaningful sustained impact, it is recommended that schools recruit a member of the school community, who is willing and has capacity to lead the initiative, as a champion.

### 4.4. Dose Received

In all schools a decline in participation in MK was seen after the first couple of weeks, suggesting a novelty effect, whereby the program’s perceived newness may have had an initial motivational effect on pupils that was not sustained over time [40]. This is consistent with the findings observed in many other school-based interventions [41]. Motivational constructs such as enjoyment are believed to be an important determinant of long-term participation [42] and likely to play a more important role in sustaining participation. It is recommended that schools emphasize fun and enjoyment of school-based running programs and developing a positive experience over and above competition or distance covered. This can be helped by communicating and reinforcing the multiple benefits of participating to pupils, particularly the more immediate short-term benefits (e.g., fun, opportunity to get some fresh air, and spend time with friends).

### 4.5. Strengths and Limitations

The strengths of this study included the use of mixed methods, measures at multiple levels (i.e., school and individual), and the collection of these data at multiple time points during implementation and at follow-up, thereby providing a more holistic understanding of the implementation context in a sample of English primary schools.

While the modest sample size may limit the generalizability of the findings, a strength of the study was the recruitment strategy employed to ensure, as far as is possible, representativeness (in terms of size, socioeconomic status, and geographic location). Due to financial limitations and the ambition to have equitable representation, the evaluation only focused on one year-group (Year 5). While this may limit the applicability of the findings across the primary school setting more broadly, the benefit of using this approach was the ability to examine within-school differences between Year 5 classes within the same school.

## 5. Conclusions

Findings suggest that MK is a simple, yet versatile, program that can be implemented in a variety of school-based environments. The program had good reach and was delivered with good fidelity, although the dose of MK (delivered and received) fluctuated. Many contextual factors, which differed between schools, influenced the quality of implementation and the frequency of delivery. While this complexity provides challenges, it may also offer opportunities to optimize implementation and inform adaptations specific to the schools’ contexts, thereby increasing the likelihood of positive outcomes and program sustainability.

## Figures and Tables

**Figure 1 children-07-00151-f001:**
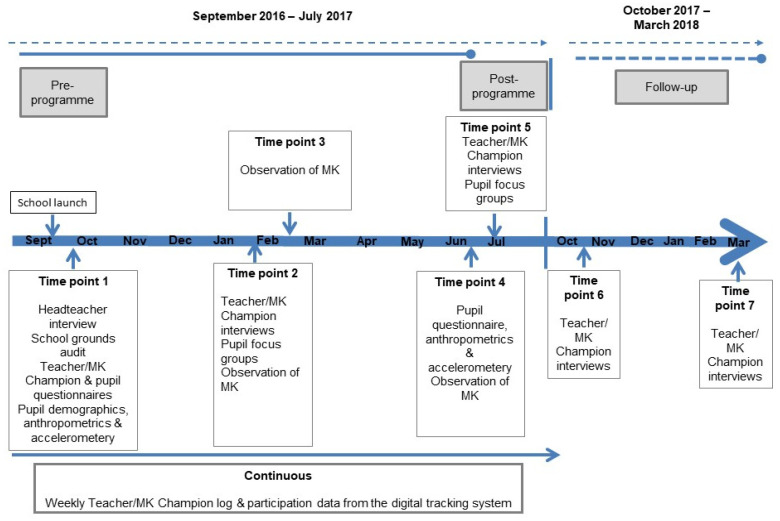
Time plan for the evaluation of Marathon Kids (MK).

**Table 1 children-07-00151-t001:** A summary of the implementation evaluation measures used.

Implementation Outcome	Implementation Outcome Definition	Description	Data Collection	
			Measure	Time Point
**Reach**	The number (%) of pupils and teachers involved in the programme and their representativeness (e.g., by BMI, SES, sex)	School level: Characteristics of schools participating in MK	Headteacher interview	Pre-programme
			Teacher questionnaire	
			Pupil questionnaire (Q-SPACE-R)	
			Observation	
		Individual level: Characteristics of teachers/staff participating in MK	Teacher questionnaire	
		Pupil: Number/percentage of pupils participating in MK	Participation (DTS)	
		Characteristics of pupils participating in MK	Demographic (SIMS) Anthropometrics Questionnaire	
**Fidelity of delivery**	The extent to which the delivery of MK was implemented as planned	School level: Conformity to the implementation strategy i.e., use of the ‘MK five pillars’ *	Teacher/MK Champion interview Pupil focus group	TP2 & TP5
		Individual level: Teacher, Pupil	Implementation log Participation (DTS)	Continuous
			Observation	TP2, TP3, TP4
**Dose delivered/received**	How much of the MK sessions were delivered and received	School level: The number of sessions/week pupils participated in	Teacher/MK Champion interview Pupil focus group	TP2 & TP5
			Implementation log Participation (DTS)	Continuous
			Observation	TP2, TP3, TP4

* ‘MK five pillars’ relate to the core principles which underpin the program: Goal setting, tracking, role modelling, celebrating, and rewards (BMI = body mass index, SES = socioeconomic status, MK = Marathon Kids, DTS = digital tracking system, SIMS = school information management system, Q-SPACE-R = questionnaire assessing school physical activity environment, TP = time point.

**Table 2 children-07-00151-t002:** Summary of school characteristics.

School	Urban/Rural Description	Status	No of Pupils in the School	Ofsted Rating	IDACI (Decile)	eFSM (%)
1	Urban	Community school	186	Good	9	34.9
2	Rural village	Church of England Academy converter	58	Good	10	11.8
3	Rural town and fringe	Church of England	501	Outstanding	6	16.4
4	Rural town and fringe	Church of England Academy converter	215	Good	4	24.7
5	Urban	Academy sponsor led	355	Requires improvement	5	18.8

(IDACI = Income Deprivation Affecting Children Index, eFSM = free school meal eligibility).

**Table 3 children-07-00151-t003:** Individual level characteristics of pupils participating in Marathon Kids.

Variable	Male (*n* = 81; 47.4%)	Female (*n* = 90; 52.6%)	Total (*n* = 171)
Age	9.6 ± 0.4	9.7 ± 0.3	9.7 ± 0.3
**Ethnicity**			
White British	52 (64.2%)	53 (58.9%)	105 (73.9%)
Asian (South Asian and East Asian)	3 (3.7%)	3 (3.3%)	6 (3.5%)
Other	14 (17.3%)	19 (21.1%)	33 (19.3%)
eFSM,	11 (13.6 %)	10 (11.1%)	21 (12.3%)
IMD decile score	7.14 (± 2.7)	6.41 (± 2.8)	6.77 (± 2.6)
Overweight and obese	25 (30.5%)	24 (24.5%)	49 (26.1%)
**Accelerometer variables**			
Monitor wear (minutes/day)	721.4 (± 83.2)	704.6 (± 83.2)	724.3 (± 71.5)
School day * monitor wear (minutes/day)	267.32 (± 122.7)	221.5 (± 114.3)	306.6 (± 104.8)
Sedentary (minutes/day)	411.6 (± 67.6)	420.9 (± 71.3)	416.83 (± 69.6)
Light (minutes/day)	239.2 (± 39.9)	229.6 (± 44.5)	233.84 (± 42.7)
MVPA (minutes/day)	68.1 (± 24.6)	51.6 (± 18.9)	58.9 (± 23.1)
Pupils achieving ≥60 min of MVPA on every valid day	35 (23.8%)	22 (14.9%)	57 (38.8%)
Pupils achieving ≥30 min of MVPA on every valid school day	16 (76.2%)	5 (23.8%)	21 (24.4%)

IMD 2015 decile scores range from 1 to 10; 1 is the least deprived and 10 is the most deprived. Note, ethnicity was based on 142 participants, weight status based on 153 participants, IMD data based on 142 participants, eFSM based on 140 participants, standing height based on 156 participants, and accelerometer data based on 96.7% of participants providing valid data. Pupils achieving ≥30 min of MVPA on every valid school day was based on 56.6% of participants providing valid data to examine school-based physical activity. (Values given are means and standard deviations unless otherwise stated; eFSM = free school meal eligibility; MVPA = moderate to vigorous physical activity; IMD = Index of Multiple Deprivation; * school day was defined as 08:30–15:00).

**Table 4 children-07-00151-t004:** Summary of the assessment of schools’ fidelity to Marathon Kids (MK).

MK Implementation Strategy	Fidelity Markers	School 1	School 2	School 3	School 4	School 5	Fidelity Marker Total
Celebration	A launch event is held	2	2	2	2	2	10
Role modelling	MK Champion and MK Ambassadors identified	2	2	2	2	2	10
Goal setting	Opportunity for pupils to participate once to twice a week	2	2	2	2	2	10
Monitoring/tracking	Takes place along an identified marked route	2	2	1	2	1	8
	Lap bands are used to monitor laps completed	2	2	1	2	2	9
	DTS used to track progress	2	2	2	2	2	10
Reward	Rewards issued at key milestones	1	1	2	1	1	6
							63
**School total:**		13 (81.3%)	13 (81.3%)	12 (75%)	13 (81.3%)	12 (75%)	

0 = not employed at all, 1 = partially employed, 2 = fully employed.

**Table 5 children-07-00151-t005:** Dose of Marathon Kids (MK) delivered (based on teacher implementation log data).

School	Class	No of Weeks of Implementation	No of Weeks of Completed Log Data	No of Weeks of Delivery	No of Sessions of MK Offered	Average Duration of MK Session Per Week *					
		(From Launch Date to Close of Programme, Minus Holidays)		(The Number of Weeks At Least 1 Session of MK Was Held)		0–5 Min	5–10 Min	10–15 Min	15–20 Min	20–30 Min	More Than 30 Min
School 1		28	23 (82%)	19 (83%)	33	0	0	0	0	6.1%	93.9%
School 2		24	24 (100%)	23 (96%)	45	0	0	0	36.6%	2.9%	0
School 3	Class A	21	19 (86%)	15 (79%)	28	3.6%	10.7%	28.6%	46.4%	10.7%	0
	Class B	21	19 (86%)	15 (79%)	29	3.4%	10.3%	20.7%	13.3%	2.9%	0
	Class C	21	19 (86%)	15 (79%)	28	3.6%	10.7%	21.4%	55.2%	10.4%	0
School 4		23	20 (87%)	18 (94.7%)	70	0	0	10%	21.7%	14.7%	0
School 5	Class A	25	20 (80%)	16 (84%)	15	5.9%	0	0	1.7%	35.4%	11.1%
	Class B	25	20 (80%)	16 (84%)	15	5.9%	0	0	1.7%	35.4%	11.1%
Range		21–28	19–24	16–23	15–70	0–1	0–3	0–8	0–49	1–18	0–31
Mean		23.5	21.2	18.2	32.9						

* Percentages do not all add up to 100 due to missing data.

**Table 6 children-07-00151-t006:** Dose of Marathon Kids (MK) received (based on digital tracking system data).

School	Class	Dose of MK Received									
		Total Distance (km)	Average Distance Per Pupil (km)	Average Distance Per Pupil Per Week (km)		Boys (%)			Girls (%)		
					Min–max	Low	Medium	High	Low	Medium	High
School 1		597.6	21.3	0.76	0.14–1.7	25	25	50	25	58.3	16.7
School 2		258.2	32.3	1.34	0.92–1.8	50	0	50	16.7	66.7	16.7
School 3	Class A	724.0	27.8	1.33	0.73–2.9	25	25	50	61.5	30.8	7.7
	Class B	447.1	26.3	1.25	0.81–1.8	20	30	50	0	66.7	33.3
	Class C	602.7	23.2	1.10	0.80–1.6	0	66.6	33.3	17.6	82.4	0
School 4		419.8	16.8	0.76	0.20–1.3	0	50	50	30.8	61.5	7.7
School 5	Class A	448	20.4	0.81	0.02–1.9	28.6	21.4	50	12.5	50	37.5
	Class B	247.4	11.8	0.47	0.02–0.9	11.1	77.8	11.1	40	60	0
Total (n)						15	28	34	24	51	10
Total (%)						19.5	36.4	44.1	28.2	60	11.8

Data based on data from 162 pupils, (km = kilometer).

**Table 7 children-07-00151-t007:** Summary of the level of implementation of Marathon Kids (MK).

School	Dose of MK Delivered	Fidelity of Delivery to MK (%)	Level of Implementation of MK
School 1	Medium	81.25	Medium
School 2	Medium	81.25	Medium
School 3	Medium	75	Medium
School 4	High	81.25	High
School 5	Low	75	Low

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
