# Peer review of "An Evaluation of the Implementation of a UK School-Based Running Program"

_children, 2020, doi:10.3390/children7100151_

Round 1

Reviewer 1 Report

This is an interesting paper which explores the complexities of school-based physical activity interventions; an area of developing interest. This is incredibly relevant for other researchers undertaking research in a school context. Due to the complexity of the various elements discussed within the paper, there is a lot to comprehend. However, the authors have done a great job of ensuring clarity for the most part.

General points to improve clarity:

  • Originally I noted that it would be nice to pull out more of the pupils' views. However, I note that you have published an additional paper detailing these aspects. It may be useful to have a reminder at the beginning of the qualitative responses that there is more information published elsewhere.
  • Line 106 - 'two from each tertile'? Would this not be 6 schools? More clarity would be beneficial here.
  • Line 109 - Similarly, the paper states that 'Five schools were recruited to the study representing a reach of 41.7%', though it mentions that the schools were selected from 32 schools?
  • Some of the methods detailed (e.g. teacher PA questionnaire) aren't referred to again after the methodology. Do they need to be included? If so, and the results are published elsewhere, this may need to be noted/referenced.
  • With fidelity being one of the main aspects, it would be useful if the individual schools scores for fidelity could be reported, perhaps against the individual characteristics. It is also not clear from Table 6 how the fidelity of delivery was calculated as a percentage, as earlier scores of 0, 1, and 2 were mentioned.
  • Line 313 - How were the average distances calculated? This is unclear.
  • Did school 4 and 5 not have an MK Champion? Why not?

Some minor typos/changes are listed below:

  • The referencing style needs updating to be consistent throughout.
  • General proofreading for spelling and punctuation required.
  • Fig 1 - Should observation TP2 come before TP3? This appears chronologically later.
  • Line 132 - Some instances refer to school-level measures and some refer to organisation-level. Continuity would improve clarity.
  • Table 2 - Decile - spelling.
  • Table 4 - For the no. of weeks of implementation, the range/mean don't appear to correspond.
  • Line 407 - The quote needs to be in italics.

Author Response

Manuscript ID: children-942783

Response to Reviewers

We thank the reviewers for their comments, which we found helpful in revising our manuscript. We have edited the manuscript based on the feedback received and provide a point-by-point response to the reviewers’ comments below.

Reviewer 1

This is an interesting paper which explores the complexities of school-based physical activity interventions; an area of developing interest. This is incredibly relevant for other researchers undertaking research in a school context. Due to the complexity of the various elements discussed within the paper, there is a lot to comprehend. However, the authors have done a great job of ensuring clarity for the most part.

We appreciate the reviewer taking the time to give feedback and we are grateful to the editors for the opportunity to revise our manuscript.

Originally I noted that it would be nice to pull out more of the pupils' views. However, I note that you have published an additional paper detailing these aspects. It may be useful to have a reminder at the beginning of the qualitative responses that there is more information published elsewhere.

Thank you for this suggestion. Reference to our earlier work on pupils’ experiences of participating in Marathon Kids has been made in line 178.

Line 106 - 'two from each tertile'? Would this not be 6 schools? More clarity would be beneficial here.

More detail relating to the recruitment of schools (and reasons for non-participation) have been provided to add clarity and also address the point raised below.

Line 109 - Similarly, the paper states that 'Five schools were recruited to the study representing a reach of 41.7%', though it mentions that the schools were selected from 32 schools?

Please see above.

Some of the methods detailed (e.g. teacher PA questionnaire) aren't referred to again after the methodology. Do they need to be included? If so, and the results are published elsewhere, this may need to be noted/referenced.

Thank you for highlighting, it is correct that section 2.4.2.1 (teacher questionnaire) is not referred to in the results and so has been removed and the subsequent subsections renumbered.

With fidelity being one of the main aspects, it would be useful if the individual schools scores for fidelity could be reported, perhaps against the individual characteristics. It is also not clear from Table 6 how the fidelity of delivery was calculated as a percentage, as earlier scores of 0, 1, and 2 were mentioned.

An additional table summarising the fidelity scores and how they were calculated has been provided (Table 4).

Line 313 - How were the average distances calculated? This is unclear.

The data is taken from the Digital Tracking system. An explanation on the DTS is provided in the programme description (lines 79-83) and emphasised in line 338.

Did school 4 and 5 not have an MK Champion? Why not?

All schools had a MK Champion for the project (line 237) but differed in terms of effectiveness. Some additional text has been provided (line 392) to add clarity.

Some minor typos/changes are listed below:

  • The referencing style needs updating to be consistent throughout.
  • General proofreading for spelling and punctuation required.
  • Fig 1 - Should observation TP2 come before TP3? This appears chronologically later.
  • Line 132 - Some instances refer to school-level measures and some refer to organisation-level. Continuity would improve clarity.
  • Table 2 - Decile - spelling.
  • Table 4 - For the no. of weeks of implementation, the range/mean don't appear to correspond.
  • Line 407 - The quote needs to be in italics.

Many thanks for highlighting these discrepancies, which have all been addressed in the revised manuscript.

Reviewer 2 Report

The paper entitled “An evaluation of the implementation of a UK school-2based running programme” is consistent with the profile of the Journal Children.

  1. The information presented in the abstract is adequate and describes thoroughly what the paper is about.

  1. The introduction explains thoroughly the scientific background. Therefore I am convinced why this context is important. The authors use up-to-date literature to present the discussed problem in the paper. Until line 75 the author uses different bibliographic system in comparison to the rest of the paper.

  1. Material and methods section is well prepared. The information provided in sections 2.1.-2.5 is clearly presented. Figure 1 lacks the tile.

  1. The outcomes are clearly described. Key results are summarized with reference to study objectives. The discussion section provides the reference to other contribution in the studied area. The conclusions are supported by the results.

Author Response

Manuscript ID: children-942783

Response to Reviewers

We thank the reviewers for their comments, which we found helpful in revising our manuscript. We have edited the manuscript based on the feedback received and provide a point-by-point response to the reviewers’ comments below.

Reviewer 2

The information presented in the abstract is adequate and describes thoroughly what the paper is about.

We appreciate the reviewer taking the time to give feedback.

The introduction explains thoroughly the scientific background. Therefore I am convinced why this context is important. The authors use up-to-date literature to present the discussed problem in the paper. Until line 75 the author uses different bibliographic system in comparison to the rest of the paper.

Thank you for highlighting this, we have now rectified the referencing to be consistent throughout.

Material and methods section is well prepared. The information provided in sections 2.1.-2.5 is clearly presented. Figure 1 lacks the tile.

Thank you for highlighting this, we have now included a title for Figure 1.

The outcomes are clearly described. Key results are summarized with reference to study objectives. The discussion section provides the reference to other contribution in the studied area. The conclusions are supported by the results.